# Understanding how and why audits work in improving the quality of hospital care: A systematic realist review

**Lisanne Hut-Mossel**[1]*, **Kees Ahaus**[2], **Gera Welker**[3], **Rijk Gans**[4]

**1** Centre of Expertise on Quality and Safety, University Medical Centre Groningen, University of Groningen, Groningen, The Netherlands, **2** Department Health Services Management & Organisation, Erasmus School of Health Policy & Management, Erasmus University, Rotterdam, The Netherlands, **3** University Medical Center Groningen, University of Groningen, Groningen, The Netherlands, **4** Department of Internal Medicine, University of Groningen, University Medical Centre Groningen, Groningen, The Netherlands

☯ These authors contributed equally to this work.
* p.a.mossel@umcg.nl

**Data Availability Statement:** All relevant data are within the paper and its Supporting information files.

**Funding:** The author(s) received no specific funding for this work.

## Abstract

### Background

Several types of audits have been used to promote quality improvement (QI) in hospital care. However, in-depth studies into the mechanisms responsible for the effectiveness of audits in a given context is scarce. We sought to understand the mechanisms and contextual factors that determine why audits might, or might not, lead to improved quality of hospital care.

### Methods

A realist review was conducted to systematically search and synthesise the literature on audits. Data from individual papers were synthesised by coding, iteratively testing and supplementing initial programme theories, and refining these theories into a set of context–mechanism–outcome configurations (CMOcs).

### Results

From our synthesis of 85 papers, seven CMOcs were identified that explain how audits work: (1) externally initiated audits create QI awareness although their impact on improvement diminishes over time; (2) a sense of urgency felt by healthcare professionals triggers engagement with an audit; (3) champions are vital for an audit to be perceived by healthcare professionals as worth the effort; (4) bottom-up initiated audits are more likely to bring about sustained change; (5) knowledge-sharing within externally mandated audits triggers participation by healthcare professionals; (6) audit data support healthcare professionals in raising issues in their dialogues with those in leadership positions; and (7) audits legitimise the provision of feedback to colleagues, which flattens the perceived hierarchy and encourages constructive collaboration.

**Competing interests:** The authors have declared that no competing interests exist.

**Abbreviations:** CMOcs, context–mechanism–outcome configurations; EPOC, Cochrane effective practice and organisation of care review group; O, Final outcome; PO, Proximal outcome; PROMs, Patient-reported outcome measures; QI, Quality improvement; QI-MQCS, Quality improvement minimum quality criteria set.

## Conclusions

This realist review has identified seven CMOcs that should be taken into account when seeking to optimise the design and usage of audits. These CMOcs can provide policy makers and practice leaders with an adequate conceptual grounding to design contextually sensitive audits in diverse settings and advance the audit research agenda for various contexts.

## PROSPERO registration

CRD42016039882.

## Introduction

In recent years, quality and safety issues have become increasingly important in hospital care following an increased focus on both clinical outcomes and patient satisfaction. Health authorities and organisations prioritise audits as a quality improvement (QI) approach by systematically evaluating delivered care, identifying areas for improvement and implementing changes for the better [1]. Several types of audits, including external audits, internal audits, peer reviews and clinical audits, have been used, but all share the problem that the implementation of suggested improvements often fails to close the quality gap they exposed [1–3].

The limited effectiveness of audits suggests that carrying out audits and implementing improvements is not a straightforward process [4–6]. Although various explanations for how audits work have been offered, there has been little in-depth theorising about the causal mechanisms that determine the effectiveness of audits in a given context [7, 8].

The audits used in the area of improving healthcare can roughly be divided into: (1) external audits, used to gain insight into a hospital's compliance with external criteria (e.g. accreditation, certification, external peer reviews); (2) internal audits, often in preparation for an external audit; and (3) clinical audits, carried out as a local initiative by healthcare professionals [1, 3] (Table 1). Although there are differences in the scope and approaches used in audits, they all share the objective of improving the quality of hospital care.

Audits are posited to increase accountability and improve the quality of hospital care through systematic monitoring and evaluation. However, many audits are designed without explicitly building on previous research or guided by theory [16–18]. As a result, there has been little progress with respect to identifying the key ingredients for successful audits. The variety in the levels of audits, together with the heterogeneity of their contexts, suggests that it is unlikely that audits work in the same way in every setting. Given this situation, a detailed understanding of the contextual factors and the causal mechanisms that influence the effectiveness of audits is necessary if one is to improve the design and optimisation of the audit process.

The aim of the current study is to understand the mechanisms and contextual factors that determine why audits might, or might not, lead to improved quality of hospital care. Two related research questions were formulated:

1. Through which mechanisms do audits deliver their intended outcomes?

2. What contextual factors determine whether the identified mechanisms result in the intended outcomes?

**Table 1. Types of audits [9] (S1 Protocol).**

| |
|---|
| **Externally driven audits**–For example, accreditation, certification and external peer reviews are strongly anchored in quality assurance (QA), referring to initiatives designed to assure compliance with minimum quality standards [10, 11]. Quality assurance is defined as: "The part of quality management focused on providing confidence that quality requirements will be fulfilled" [11]. External audits are used to assess the quality system of a healthcare organization based on specified standards and are conducted by external auditors [12]. |
| **Internal audits**–This type of audit is conducted by internal auditors of the hospital's own organisation, such as quality officers or healthcare professionals from another department than the one being audited to guarantee independent judgement. Internal audits are used to evaluate the quality system based on standards as well. They are conducted to prepare for external audits. Healthcare organisations also use internal audits to continuously improve the quality of healthcare. Internal audits are designed to evaluate and improve the effectiveness of the organisation's quality management system and focus more on organisational conditions than on performance of healthcare professionals and patient outcomes [3]. |
| **Clinical audits**–Clinical audits differ from other types of audits in that they are mostly initiated and undertaken by healthcare professionals. Clinical audits represent a shift from QA to QI, with a focus on increasing the ability to fulfil quality requirements, seeking to improve care, enhance performance and prevent poor care [11, 13]. This process takes place continuously as part of everyday routines [1, 12, 13]: healthcare professionals work together to collect data and evaluate their own practices. Following this, they intend to develop and apply improvements in daily practice, and then the audit cycle is repeated to demonstrate improved and sustained improvements [14]. Hence, clinical audits do not necessarily use external criteria and are not carried out in response to external demands as the initiative comes from the healthcare professionals themselves [15]. |

## Methods

We adopted a realist review approach to address our research questions. We were guided by the RAMESES publication standards for realist reviews and we followed the PRISMA guidelines for systematic searching of the literature (S3 Table) [19, 20].

In addition to the approach of a systematic review, we have chosen to use a realist review approach because this permits us to understand in what circumstances and through what processes audits might, or might not, lead to improved quality of hospital care and why. This approach recognises that the success of audits is shaped by the way in which they are implemented and the contexts in which it is implemented. Realist reviews belong to the school of theory-driven inquiry and are concerned with how an intervention works, rather than focussing solely on whether an intervention works. Furthermore, the realist review methodology is specifically designed to cope with the complexity and heterogeneity (e.g. in a study's design and context [21]), identified in previous research on audits [5, 9, 22, 23]. Table 2 provides definitions of realist concepts. Realist reviews recognise that interventions are complex and can rarely be delivered consistently due to differences in context [19]. The unit of analysis in this realist review was not the audit itself, but the programme theories about how, why and in what circumstances audits might work. Thus, the focus of this review is on mechanisms that affect change and not on the type of audit conducted. A realist review examines the interaction between an intervention, the context (C) or 'setting' in which it is applied, the mechanism that describes how the participants use the intervention's resources ($M_{resource}$) and how they respond to these resources ($M_{reasoning}$), and the intended or unintended outcome (O) in a set of primary studies [24, 25]. $M_{resource}$ and $M_{reasoning}$ together constitute of a mechanism, but explicitly disaggregating them helps in operationalising the difference between the intervention strategy and the mechanisms. As recalled by Pawson and Tilley (2004) mechanisms are linked to, but not synonymous with the intervention strategy [24]. Mechanisms and the intervention strategy are located at different levels of abstraction [26]. The attributes of the intervention refer to strategies and implemented activities whereas the attributes of mechanisms are centred on the elements of individual or collective reasoning or reactions of participants in regard of the available resources offered by the intervention (Table 2) [24–27]. As Dalkin et al.

**Table 2. Glossary of terms.**

**Realist review**–Is a theory-driven approach to synthesising quantitative, qualitative or mixed methods research, from a perspective based in Realism. One of the key tenets of realist methodology is that programmes work differently in different contexts–hence an audit that achieves 'success' in one setting may 'fail' (or only partially succeed) in another setting, because the mechanisms needed for success are triggered to different degrees in different contexts [25, 28]. A realist review does not provide a definite answer as to whether 'something works or not'; rather it answers questions of the general format 'what works for whom under what circumstances, how and why?' to enable informed choices about further use and/or research. The results of a realist explanation are context-mechanism-outcome configurations (CMOcs) constituting a refined programme theory. The refined theory can then be tested in a subsequent realist evaluation covering the same kind of project or programme.

**Context**–Context refers to the 'setting' in which interventions take place. As conditions change over time while the audit is executed, the context may change as well and reflects aspects of that change in conditions. As such, contextual elements influence the relationship between audits and their outcomes and, vice versa, the process of the audit and its outcome will influence the context (for example, the outcomes of an audits may generate a culture change) [24, 25].

**Mechanism**–Mechanisms are a combination of resources offered by the intervention ($M_{resource}$ e.g. information, skills or support) and the participants' response to these resources ($M_{reasoning}$). Intervention resources are introduced in a context that might affect participants behaviour based upon their beliefs, attitudes or logic (reasoning), which will, in turn, affect the outcome [24, 25, 27].

**Outcome**–Outcomes can be either intended (did the project succeed against the criteria it set itself at the outset), or unintended, and can be proximal or final. Proximal outcomes can be changes in skills, commitment or intentions of healthcare professionals, whereas final outcomes relate to improvements in quality of care.

**Context-mechanism-outcome configuration (CMOc)**–In realist reviews, observed associations are expressed as context-mechanisms-outcome (CMO) configurations, to explain how particular contexts trigger mechanisms to generate certain outcomes [24]. CMO configurations can be applied to design future audits across different contexts.

**Programme theory**–Programme theories are theory informed associations organised as 'abstracted descriptions' about the ideas and assumptions underlying how, why and in what circumstances complex social interventions work [19]. Typically, realist reviews start with initial programme theories (in our review formulated as 'If-then statements') and end with a more refined programme theory, expressed as CMOcs.

put it, "resources must be introduced into a pre-existing context, which in collaboration induces an individual's reasoning, leading to an outcome." [27] (p.5). Data were collected and combined to identify context–mechanism–outcome configurations (CMOcs).

The protocol for this research, with a detailed description of the research team composition, search process and selection of primary studies, has been previously published (S1 Protocol) [9]. The first step of our synthesis sought to identify initial programme theories and was guided by the question 'What is the intrinsic logic of audits?' i.e., what explains why audits are assumed to be a good idea [19]. Drawing on the literature on effectiveness of QI strategies, we developed several initial programme theories in the form of 'if-then' statements (Table 3) [19, 29]. Next, a systematic literature search was conducted in MEDLINE, Embase, PsycINFO, Academic Search Premier, Business Source Premier, Emerald Insight, Cochrane Library and Web of Science covering the period 2005 –August 2020 (S1 Box). No changes were made to the inclusion criteria as formulated in the review protocol (S1 Protocol) [9]. Eligible studies were empirical studies that evaluated the effects of audits in hospital settings within high-income countries, and no restrictions were placed on the type of study design (Box 1). Articles were excluded if they did not meet the inclusion criteria [9]. Furthermore, articles reported in languages other than English were excluded to avoid misinterpretation of the content of an article due to language barriers.

Each paper's quality was appraised by two reviewers (LH plus KA, GW or RG). To determine its rigour, the quality of the evidence in each paper was presented in the form of an evidence-level table following the criteria established by the Cochrane 'Effective Practice and Organisation of Care' (EPOC) review group [30]. In addition, the 'Quality Improvement

**Table 3. Key findings.** Initial programme theory in the form of 'If-Then' statements in relation to the CMO configurations (observed associations), key quotations and reflections of the focus group meeting.

| Initial programme theory | CMO configuration | Key quotations from the included studies | Reflections of the focus group meeting |
|---|---|---|---|
| **Theory 1**. *If* the organisation has appropriate systems and skills in place to monitor, track and evaluate the results of an audit (both anticipated and unanticipated) *then* the improvements are more likely to be incorporated and sustained [120]. | **CMOc1. Externally initiated audits create QI awareness although their impact on improvement diminishes over time** | "Once the on-site survey concluded, improvements continued, but the rate levelled off. Essentially, the positive trend prior to accreditation increased during accreditation and continued post-accreditation, but it began to plateau." [51] (p.5) | The focus group confirmed this pattern. The focus group noted that external audits are often experienced as an imposed way to comply with external requirements. The group confirmed that the number of years that an organisation participated in external audits does negatively affect the extent of changes that take place; they had the experience that, after about three years from initiating the audit, the rate of improvements tailed off: *"The audit has, especially at the start, considerable effect. The conversations are in fact new, and they [healthcare professionals and organisational members] talked for the first time with someone [the auditor] who could help them to have a closer look at how they are performing and this person could help them further."* [Participant 1] |
| **Theory 2**. *If* an audit focuses on continuous improvement rather than on assurance and compliance *then* the audit is more proactive and facilitates change [13, 121]. | In a context where the audit is mandated by an external party (C), organisational members are instructed to participate in the audit process and collect data for the audit to be able to comply with the requirements (M$_{resource}$). Although these audit activities are imposed, the interest and awareness of healthcare professionals and organisational members in QI are raised (M$_{reasoning}$), even after the external pressure has diminished (C). After a couple of years, the audit no longer provides a challenge to healthcare professionals and organisational members to improve quality (M$_{reasoning}$) and the rate of improvement levels off (PO). | | |
| **Theory 3**. *If* healthcare professionals are actively involved in audit processes and recognise the need for improvement *then* they identify themselves with the findings of an audit which makes them more likely to accept the findings and, where appropriate, implement change [5, 122, 123]. | **CMOc2. A sense of urgency felt by healthcare professionals triggers engagement with an audit** | "The present inquiry began when the senior author reviewed a day in the OR on which cases started late, families expressed frustration with long preoperative stays at the hospital for short procedures, the OR finished later than scheduled, and little operation had occurred because the room was often not occupied by a patient." [82] (p.783) | CMO configuration not discussed during the focus group meeting |
| | In a context where an urgency to improve quality is present (C), healthcare professionals undertake actions and initiate local audits as an improvement instrument (M$_{resource}$), are willing to participate in the audit as they feel engaged (M$_{reasoning}$), and recognise the need for improvement (M$_{reasoning}$), thereby increasing the likelihood of actual improvements in patient care (PO). | "The organizational focus remained to be a source of frustration. Twenty interviewees expressed their concerns on having to repeat all the organizational items in the self-review and site visit in a fourth participation round. As a result, only 12 persons would still find a fourth participation worthwhile without major changes in the program." [106] (p.483) | |
| **Theory 4**. *If* the driving forces behind an audit are champions who advocate the audit *then* this increases the willingness of healthcare professionals to adapt and accept suggested improvements, which results in greater adoption of the improvements by individual healthcare professionals in an organisation [7, 120]. | **CMOc3. Champions are vital for an audit to be perceived by healthcare professionals as worth the effort** | "A group of internal medicine residents, led by a resident champion, considered several potential quality goals (. . .). A retrospective chart review of 100 randomly selected patients discharged from the medical service in June 2009 revealed that communication with primary care physicians was documented in only 55% of cases. Therefore, the residents challenged their colleagues to increase the rate of documented primary care physicians contact to 80% or higher." [69] (p.473) | CMO configuration not discussed during the focus group meeting |
| **Theory 5**. *If* an organisation has leadership committed to quality of care *then* this positively influences the organisational culture and thus the implementation of quality improvement measures [7]. | A healthcare professional who is widely recognised as a champion (C) and who is adequately supported by the organisation (C) acts as a role model for the espoused change (M$_{resource}$) and challenges colleagues to implement quality improvements in practice (M$_{resource}$). Healthcare professionals consider the audit relevant since one of their peers is the driving force, and this fosters trust (M$_{reasoning}$). This results in increased attention to the quality of delivered care (PO) and a stronger commitment to QI by healthcare professionals (PO). | | |

*(Continued)*

**Table 3.** (Continued)

| Initial programme theory | CMO configuration | Key quotations from the included studies | Reflections of the focus group meeting |
|---|---|---|---|
| **Theory 6**. *If* an audit is initiated in a bottom-up way by healthcare professionals *then* this allows ownership and autonomy, which fosters trust, goodwill and confidence such that implementation and sustained surveillance of the suggested improvements are more likely to occur [124]. | **CMOc4. Bottom-up initiated audits are more likely to bring about sustained change**<br><br>In a context where healthcare professionals work closely together, respect each other's competences and are sensitive to each other's work and needs (C), they are able to define quality improvements ($M_{resource}$) that are recognised as valuable by all healthcare professionals involved ($M_{reasoning}$), ensuring appropriateness and fostering acceptance of sustainable improvements in practice ($M_{reasoning}$). Increased and improved communication between healthcare professionals (PO) favours reflection and feedback ($M_{reasoning}$) that will result in improved multidisciplinary care collaboration (PO). | "A clearer job description, better communication between professionals and more unambiguous agreements on how to arrange postoperative care were mentioned as being important subjective outcome measures. These logistic alterations and specific arrangements on how a professional would provide care or information to the patient, are difficult aspects to evaluate by objective (hard) end-points, but improvement in this area is of great importance when improving quality of care is the main goal." [80] (p.275) | The focus group acknowledged the importance of respecting each other's competences and being sensitive to each other's work and needs by healthcare professionals. They also recognised improved reflection, feedback and collaboration as an outcome: *"Regarding clinical audits that are performed within professional groups, I noted that these reflections strengthened the understanding among professionals because they can discuss with each other the outcomes of the audit and, as a result, this boosts solidarity"* [Participant 2] |
| No initial programme theory | **CMOc5. Knowledge-sharing within externally mandated audits triggers participation by healthcare professionals**<br><br>In a context in which the audit is mandated by an external party (C), healthcare professionals are brought together to exchange ideas and knowledge ($M_{resource}$), which makes them to see the audit as a learning opportunity ($M_{reasoning}$), thus increasing their willingness to participate ($M_{reasoning}$). This will result in better understanding of the challenges facing the organisation (PO), changes being more quickly implemented and spread throughout the organisation (PO) and improved communication within and between teams (PO). | "At the beginning of the self-assessment phase, staff seated around the table had divided into three groups, each of which spoke to the moderator but not to the other groups. By the end of the self-assessment phase, staff from different sites [of the organisation] sat in mixed groups around the table. They also exchanged protocols, discussed means of implementing common working procedures, and collaborated on better integrating the patient pathway within the organization." [83] (p.8) | The focus group acknowledged the importance of knowledge sharing among healthcare professionals. The clinical relevance of the audit topic was an important factor for healthcare professionals to decide to participate: *"What do we need in order to deliver high-quality care, good education and to conduct good research? If that's your starting point, then everyone is willing to contribute."* [Participant 3] |
| No initial programme theory | **CMOc6. Audit data support healthcare professionals in raising issues in their dialogues with those in leadership positions**<br><br>In a context of a safe organisational culture (C), healthcare professionals are able to substantiate their requests for changes using audit data ($M_{resource}$) and feel strengthened in their dialogue with organisational leaders to convince them of the need for change in practice ($M_{reasoning}$). These requests for changes by healthcare professionals will result in organisational leaders adopting a more active approach to QI (PO). | "Not only did the accreditation recommendations cause management to adjust and modify many practices, staff also used them to convince management and the Board of Directors to adopt particular measures." [83] (p.9) | The focus group recognised that, within a safe organisational culture, healthcare professionals feel free to raise their concerns: *"I think that if there's a safe culture, that they [healthcare professionals] actually feel empowered, and that they dare to share issues and to discuss data of audits with leaders."* [Participant 4] |
| No initial programme theory | **CMOc7. Audits legitimise the provision of feedback to colleagues, which flattens the perceived hierarchy and encourages constructive collaboration**<br><br>In a context of a safe learning culture (C), healthcare professionals are encouraged to give feedback to each other during the audit ($M_{resource}$). Professionals feel that a specific audit finding justifies ($M_{reasoning}$) and empowers them in giving feedback to other staff for non-adherence ($M_{reasoning}$). Providing feedback flattens the existing perceived hierarchy (PO) and results in better team collaboration (PO). | "I would never have told a MD [Medical Doctor] to wash his or her hands without this [hand hygiene tool]" and "It [the audit] gave me justification to tell others to follow [hand hygiene] policy." [72] (p.52) | The focus group agreed that a safe learning climate is a prerequisite if healthcare professionals are to address each other's performance. In their view, the audit can be an instrument that facilitates an open discussion of quality of care and creates greater awareness of a certain policy: *"If you normally feel you are not in a position to say something about a certain subject, then the audit has the effect of empowering you to do something about it. (...) The audit facilitates discussing issues that otherwise would not have occurred."* [Participant 3] |

Minimum Quality Criteria Set' (QI-MQCS) was used to assess the completeness of the reporting of each paper [31]. To date, cut-off scores for what may be considered low, medium or high quality have not yet been determined. In order to interpret the QI-MQCS score, we used the criteria previously used by Kampstra et al. [32]: a study was weighed to be of perfect quality

> ## Box 1. Inclusion criteria
>
> Studies on accreditation, certification, peer review/Dutch visitate model (external)[a]
>
> or local clinical audit (internal)
>
> Hospital setting[b]
>
> High-income country
>
> Published in English
>
> English abstract available
>
> Description of the medical or technical content
>
> Descript of the process of how the audit was conducted
>
> Description of the impact of audit on medical and/or process outcomes
>
> *[a] This is a doctors-led and -owned system of peer review designed to assess the quality of care provided by groups of hospital-based medical specialists. Practices are surveyed every 3–5 years by a group of peers.*
>
> *[b] The rationale for focusing on hospital care is that these organisations share challenges regarding safety, effectiveness and values. In addition, these organizations can be quite similar in organisational structure.*

if > 15 items ranked *yes*, good quality if > 12 items ranked *yes*, moderate quality if > 9 items ranked *yes*) and insufficient quality if ≤9 items ranked *yes*.

Although the process of analysis and synthesis is reported in two stages, in practice it was an iterative process guided by the review questions. The key analytical process in a realist review involves iterative testing and refinement of the initial programme theories using empirical findings in data sources [28]. This process was informed by the realist synthesis approach described by Rycroft-Malone et al. [28]. During the first stage of the data extraction and synthesis process, sections of text relating to context, $M_{resource}$, $M_{reasoning}$ and produced outcomes were extracted (Fig 1). Four articles were independently coded for C, $M_{resource}$, $M_{reasoning}$ and O by each member of the research team. The results of this exercise achieved a Krippendorff's α of 0.79, which is considered an acceptable level of consistency [33, 34]. To further ensure consistent judgement, two reviewers independently coded and discussed the first 12 articles, resolving differences by consensus. This coding process was both deductive (codes based on the initial programme theories) and inductive (codes identified during data extraction). Thereafter, the remaining 73 papers were coded by the first author with the aim of confirming, adjusting, supplementing or refuting our initial programme theories. Details of the coding list for this review are provided in S2 Table. During the second stage, we compared and contrasted the evidence to identify patterns in the mechanisms across different contexts that related to diverse outcomes. The research team regularly discussed the patterns that emerged and their compatibility with the initial programme theories. Eventually, this iterative process allowed us to refine and advance the initial programme theories into a set of CMOcs that provide an explanatory account of how audits might, or might not, lead to improved quality of hospital care.

**Data Synthesis stage**

**Example**

**Stage 1. Extract data into evidence tables**
For each paper, quality was appraised and general characteristics were extracted and discussed by the research team so that data are not simply categorised but are used to begin to develop a reasoning that provides input to the synthesis.

**Data extraction sheets:**
- Mechanisms: Audit top-down or bottom-up initiated; involvement of healthcare professionals
- Contextual factors: Motivation/sense of urgency for conducting the audit
- Outcomes: Changes as a result of audit activities

**Stage 2. Code individual papers**
Four articles were individually coded for C, M (resource and reasoning) and O by the research team. Two reviewers independently coded 12 articles for C, M and O to ensure consistency of judgement.

"The present inquiry began when the senior author reviewed a day in the OR on which cases started late, families expressed frustration with long preoperative stays at the hospital for short procedures, the OR finished later than scheduled, and little operating had occurred because the room was often not occupied by a patient (Context)."[66] (p783)

"The design and implementation of the study involved close collaboration between the research team and NICU personnel (Mechanism$_{Resource}$)." [71] (p.288)

**Stage 3. Identify patterns based on the codes**
Mechanisms were compared across different contexts to assess whether they consistently produced similar outcomes and were discussed within the research team. Patterns were built on the 'if-then' statements.

**Initial programme theory:** If healthcare professionals are actively involved in audit processes (M$_{resource}$) and recognise the need for improvement (C) then they identify themselves with the findings of an audit (M$_{reasoning}$) which makes them more likely to accept the findings (M$_{reasoning}$) and, where appropriate, implement change (PO).

**Patterns in codes:**
Mechanism$_{Resource}$: Audit conducted by healthcare professionals
Mechanism$_{Reasoning}$: (No) engagement
Outcome: Increased commitment to quality
Context: Deficiencies in quality of care

**Stage 4. Test, refine and supplement initial programme theory across papers**
Stage 2 and 3 were repeated by the first author. Codebook deductively applied to remaining papers. Initial programme theories were refined, supplemented or new statements were formed through discussions within the research team.

The initial programme theory was tested in other papers. In total, 21 different papers reported about (the lack of) sense of urgency and engagement of healthcare professionals. We found in several papers that a sense of urgency felt by healthcare professionals is an important condition before engagement can take place, recognising the need for improvement:
"Perhaps the most important factor in the acceptance of the audit process was the immediate realization by clinical staff that the audits were identifying major remediable gaps in performance." [71] (p.288)
"While this is unacceptable, it may be that (…) the clerking doctor considered screening for dementia 'a side line' rather than part of the contributing factors leading to admission." [61] (p.701)

**Stage 5. Refine initial programme theory into CMO configurations**
We iteratively formulated and refined potentially suitable CMOcs, returning to the coded papers from Stage 1 as necessary, until agreed by the entire research team. Recognisability of the CMOcs were discussed during a focus group.

**CMOc2. A sense of urgency felt by healthcare professionals triggers engagement with an audit** In a context where an urgency to improve quality is present (C), healthcare professionals undertake actions and initiate local audits as an improvement instrument (Mresource), are willing to participate in the audit as they feel engaged (Mreasoning), and recognise the need for improvement (Mreasoning), thereby increasing the likelihood of actual improvements in patient care (PO).

**Fig 1. Data synthesis process.**

Pawson et al. [21] argue that stakeholders should be involved in both the process of confirming the emerging findings and in dissemination activities. To that end, the recognisability of the CMOcs were discussed during a focus group meeting, which was led by an independent and experienced moderator. The nine participants were purposively chosen based on their relevant experience with audits. In this respect, the Dutch Law on Medical Research Involving Human Subjects (WMO) did not require us to seek ethical approval as the focus group would not promote clinical medical knowledge, and there was no participation by patients or use of patients' data. Informed consent was received from all participants.

## Results

Of the 13709 potentially relevant records, 85 primary papers met our eligibility criteria (Box 1). During the full-text screening stage, we had to exclude more than 400 papers, often because the audit process was not sufficiently described to be able to identify mechanisms or because the outcomes were lacking (Fig 2). Of the included papers, 61 focussed on clinical audits, 17 on accreditation/certification and 7 on peer reviews. In terms of the evidence levels established by EPOC review group [30], one randomised trial was found [35], two controlled studies [36, 37], 66 case studies [38–103], and 16 descriptive studies [104–119]. Using the QI-MQCS, the completeness of reporting scores ranged from 4 to 16 (16 being the highest possible score) (S1 Table). Of the 85 included papers, 41 scored more than 12 items *yes* and where thus considered to be of good quality. 30 articles scored poorly on the QI-MQCS with a score $\leq 9$, which is ranked as low quality. Only 61 of the 85 papers met the criteria for domain 12 (organisational readiness), that describes the QI culture and resources present in the organisation, which helps to assess the transferability of results. In addition, 52 of the 85 papers met the sustainability domain by including reference to organisational resources and policy changed needed to sustain the audit results after withdrawal of study personnel and resources. The synthesis began with evidence from the most rigorous studies ($> 12$ items ranked *yes* according to the QI-MQCS), as these studies provided the most relevant and detailed evidence about C, $M_{resource}$, $M_{reasoning}$ or O. Less rigorous studies were used to test the emerging CMOcs, but also to test alternative explanations. The findings are organised across seven CMOcs describing how audits might work to improve the quality of hospital care. These are presented below and described in more detail in Table 3. These explanatory CMO configurations were constructed by the research team based on the realist analysis. The inputs from the focus group did not materially change the CMO configurations but triangulated and enriched the literature findings.

### CMOc1. Externally initiated audits create QI awareness although their impact on improvement diminishes over time [37, 39, 51, 52, 68, 73, 83, 93, 102, 104, 110, 112, 114, 117]

Of the 17 studies that addressed externally mandated audits, eight reported on the ongoing improvement process after the initial audit had taken place. Although externally initiated audits do stimulate hospitals to improve their quality of healthcare [39, 44, 51, 68, 83, 93, 104], there seems to be a tendency towards "complacency with past improvements" and a diminishing urgency for continued QI [68, 73, 83, 104, 117]. Similarly, it seems that the rate of improvement levels off once the external pressure diminishes [51]. Several authors argue that a hospital's focus on QI was influenced by perceived external pressure: if this pressure diminished, the healthcare professionals' and organisational members' motivation also eased [51, 73].

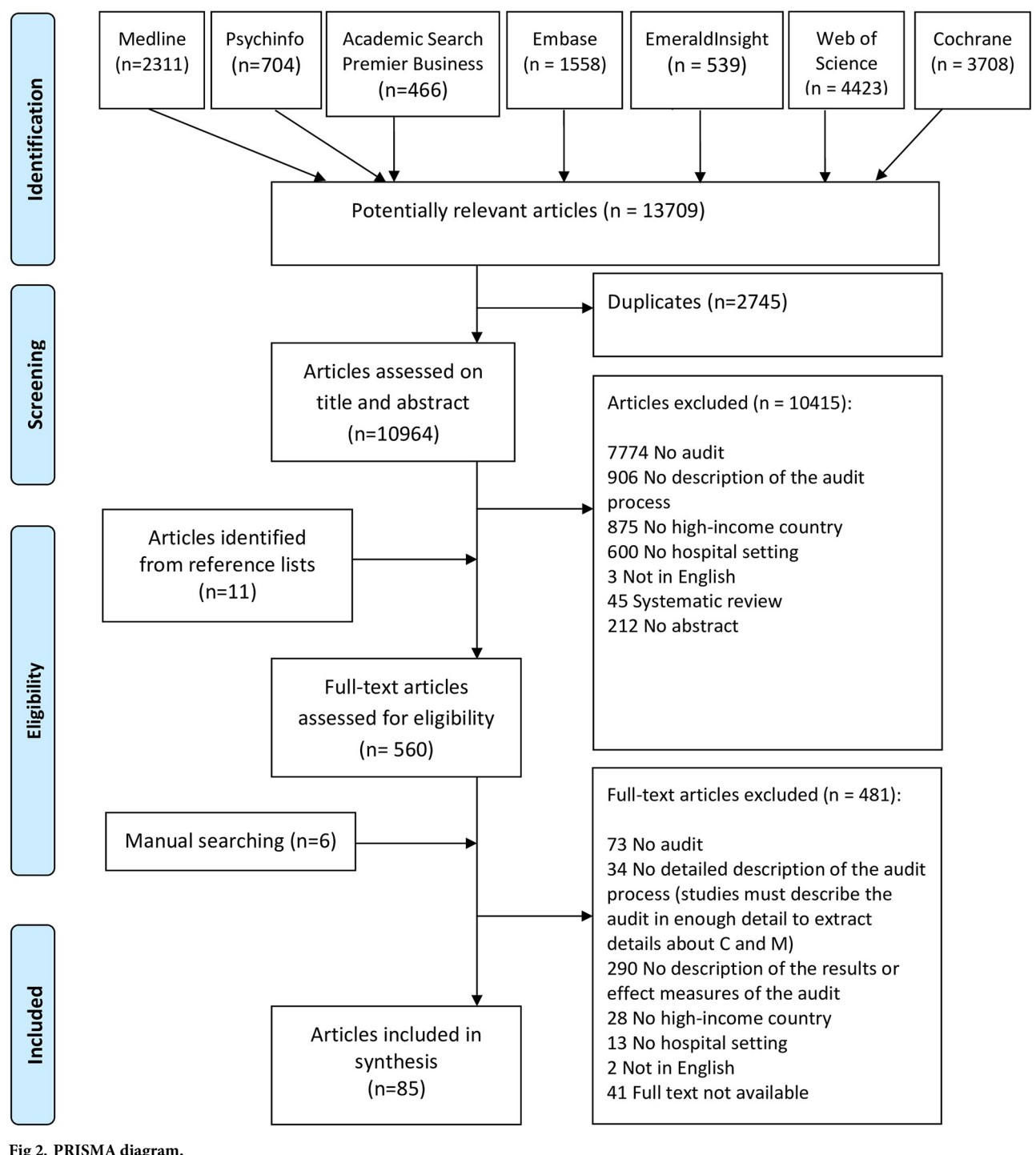

**Fig 2. PRISMA diagram.**

The number of years that an organisation had participated in audits affected the extent of the improvements that took place [73, 83, 104]. When a hospital started with audit activities, the process was perceived as a QI endeavour and as a challenge as the organisation attempted to align its care activities with best practices. However, after a few years, the extent of improvement reached a plateau and healthcare professionals and organisational members no longer

considered the audit as a driver of change [83, 93, 104]. Notwithstanding these observations, the fact that organisational members had to comply with the audit requirements, had to participate in the audit process and had to collect data for the audit, fuelled the awareness and interest of healthcare professionals and organisational members in QI [73, 83].

## CMOc2. A sense of urgency felt by healthcare professionals triggers engagement with an audit [36, 38, 40–42, 44, 45, 48, 50, 53–56, 58, 61–65, 69, 70, 75, 77, 79, 81–84, 87, 88, 93, 97–100, 102–109, 113, 117, 118]

Having a sense of urgency seems to be an important precondition for healthcare professionals to engage: if healthcare professionals perceive the current situation as untenable for themselves or for the safety of patients, this will urge them to take action [82, 105]. Several studies have described local audits that were started by intrinsically motivated healthcare professionals on issues they felt to impact on patient care [48, 53, 61, 63, 69, 82, 88, 97, 99, 103]. In contrast, if healthcare professionals perceive an audit as a 'side-line' or 'mandatory exercise', rather than a process to contribute to improved quality of care, they will not feel engaged and not put significant effort into the change [77, 118]. This is especially true when audits are externally initiated and have a strong organisational focus, sometimes without links to clinical care [102, 104, 106–108, 117]. Additionally, some studies showed that healthcare professionals reported reduced time for patient care when preparing for an external audit [38, 62, 93, 117]. Other QI initiatives, which they perceived as more relevant to patient care, were sometimes paused or stopped during the external audit [83, 117]. It was also noted that at times issues might arise during an external audit that were not in line with healthcare professionals' priorities for QI [102, 107].

## CMOc3. Champions are vital for an audit to be perceived by healthcare professionals as worth the effort [36, 43, 47, 49, 53, 54, 57, 59, 69, 74, 82, 85, 86, 89–92, 95, 101, 103, 104, 109, 119]

Champions are those healthcare professionals who are committed to implementing change, act as a role model for the intended change and are enthusiastic in convincing others in the organisation of the necessity of change [69, 82, 86, 89, 101, 103, 109]. Two kinds of champions were identified, those that initiated an audit [36, 49, 54, 59, 69, 74, 82, 89] and those nominated by the leadership of an organisation and given responsibility for carrying the audit forward [43, 47, 53, 57, 85, 86, 90, 91, 95, 101, 103, 104, 109, 119]. In successful audits champions challenge, educate and lead their colleagues to improve the quality of care [69, 90, 91, 95, 101, 119]. Furthermore, healthcare professionals consider an audit more relevant if one of their peers is the driving force, and this fosters trust [36]. This encouraged a more critical attitude and attention to the quality of the care they delivered, increases their commitment to QI [36, 69, 82, 89], and lead to changes intended to improve care being successfully implemented [36, 53, 57, 59]. A supportive organisational culture is an important contextual factor for champions to be able to perform [36, 54, 69, 86, 89, 90, 103].

## CMOc4. Bottom-up initiated audits are more likely to bring about sustained change [36, 48, 53, 57, 59, 60, 63, 67, 76, 80, 82, 91, 96]

In bottom-up initiated audits, healthcare professionals take the lead. These healthcare professionals work closely together to define appropriate improvements for their own local situation, and this ownership helps developing informal social ties and ensuring the acceptance of sustainable improvements in practice [53, 57, 59, 60, 80, 91]. For example, during an audit about

medication registration errors, nursing teams were empowered to develop their own strategies to improve practice [91]. Healthcare professionals are able to collaborate to improve the quality of care because they respect each other's competences and are sensitive to each other's work and needs [53, 57, 59]. Moreover, working together during an audit improves communication, reflection, feedback and collaboration among professionals and this encourages to think and work as a cohesive group in search for solutions on how to arrange and deliver multidisciplinary care [57, 59, 80]. As such, bottom-up initiated audits are more 'natural' and eventually more meaningful and appropriate for the local situation.

## CMOc5. Knowledge-sharing within externally mandated audits triggers participation by healthcare professionals [35, 59, 66, 73, 80, 83, 88, 106, 116]

In contexts where an audit is initiated by an external party, healthcare professionals are often obliged to work together in a working group. Several studies describe how healthcare professionals see external audits as learning opportunities as they are able to exchange ideas and knowledge [35, 73, 83, 88, 106]. Healthcare professionals find it rewarding to share knowledge about quality and care processes in working groups, and this results in a better understanding of the challenges facing the organisation. As a consequence, changes tend to be more quickly implemented and spread throughout the organisation [73]. In this way, working groups act as forums for knowledge exchange, helping to bring together different professionals or various parts of an organisation [83]. Working groups have also been shown to have other positive impacts such as, boosting the cohesion of medical and paramedical teams through better communication within and between teams [59, 66, 106, 116].

## CMOc6. Audit data support healthcare professionals in raising issues in their dialogues with those in leadership positions [63, 71, 82, 83, 88]

Audit data enable healthcare professionals to identify shortcomings in their local patient care and strengthen their confidence in discussing requests for changes with "those in positions of leadership" [82]. Even, when an audit is mandated by an external party, healthcare professionals will use the information provided by the audit to convince management of the need for change [83].

Healthcare professionals tend to express opportunities for quality improvement initiatives in contexts that are felt safe, as in a culture with a focus on collective learning and improvement, as opposed to a culture where speaking up is suppressed and mistakes are punished [63, 71, 82, 83]. In addition, by presenting audit data to leaders, healthcare professionals got the opportunity to become involved in the governance of their hospital [88]. This has enabled healthcare professionals and their leaders to share responsibility for allocating sufficient resources to improve care within the overall budgetary limits of the organisation [63, 71, 82, 83].

## CMOc7. Audits legitimise the provision of feedback to colleagues, which flattens the perceived hierarchy and encourages constructive collaboration [46, 61, 72, 94, 111, 115]

Audits create feedback opportunities in multidisciplinary teams and make it easier to have conversations among professionals from different backgrounds. Based on shared decisions over the healthcare processes to be audited, professionals feel legitimised in giving each other feedback: they feel justified and empowered in challenging each other, including professionals

who are perceived as higher up the hierarchy [72]. In some cases, lack of improvement was a result of dissonant relationships between various departments as they are often unaware of each other's duties and responsibilities [94]. The audit process thus flattens the perceived hierarchy and encourages constructive collaboration [46, 61, 94, 115]. It should be noted that this mechanism of a flattened perceived hierarchy only 'works' in a context where the learning culture is considered safe. In a context where career progression depends on hierarchical power relationships, with little room for mistakes, healthcare professionals (especially junior doctors) did not feel engaged in the audit as they perceived it to be a threat to their career prospects [111].

## Discussion

Undertaking a realist review allowed us to identify seven CMOcs that highlighted a range of enabling and constraining contextual factors and mechanisms, that are fundamental to the ways in which audits can improve the quality of hospital care. This realist review indicates that externally initiated audits create QI awareness and that knowledge-sharing within these audits is important as it triggers the participation of healthcare professionals. However, bottom-up initiated audits are more likely to bring about sustained change. A sense of urgency felt by healthcare professionals triggers engagement with an audit. Also, this review pointed out that champions are vital for audits to be perceived as worth the effort they involve. In addition, an audit can be an instrument that encourages healthcare professionals to provide each other with feedback, as well as to raise issues in dialogue with leaders and become engaged in the governance of the organisation.

### The audit as a platform to raise issues and provide feedback

From a social perspective, an audit can be viewed as a platform for the internal bonding of individuals. Bonding has been shown to be key in building a sense of 'community' that contributes to implementation effectiveness [120, 125, 126]. Here, audit meetings provide opportunities for interaction, discussion and the sharing of ideas about changing practices, and for improved communication within and between teams.

Importantly, an audit creates opportunities for feedback on multidisciplinary issues that affect all professionals involved in the care process. Studies of collaboration within multidisciplinary teams have highlighted that the medical profession remains dominant, even when there have been acknowledged attempts to "democratise" teams [127–129]. Nurses might witness and experience a variety of problems, but do not generally communicate these to medical specialists perceived to be higher up the hierarchy [130]. This study, however, has been able to demonstrate that healthcare professionals were empowered by the audit to challenge their peers and also staff who are perceived as more senior or higher up the hierarchy. Collaborative learning and speaking up to others in the hierarchy does not occur naturally in healthcare, despite its importance for improving care delivery. Therefore, creating a safe and open audit environment in which all professionals feel safe to provide feedback to each other, is of utmost importance. Hence, audits might be a next step to resolve a widespread issue in healthcare related to communication and collaboration between nurses and doctors.

Our findings regarding the importance of champions resonate with and adds to the findings of a recent systematic review and meta-synthesis by Brown et al. on the functioning of clinical performance feedback [131]. Their review reported that feedback from a clinical supervisor is likely to be perceived as having greater knowledge and skill, and is therefore more likely to be accepted [131]. Furthermore, healthcare professionals consider an audit and its outcomes as relevant for adoption in practice because one of their peers was the driving force.

It has also been suggested that the audit process creates networks of like-minded collaborators across the organisation [132, 133]. By contrast, this emphasizes the role of peer involvement and collaborative practice over the role of a clinical supervisor.

### The audit as a platform for continuing professional development by critical contributions of healthcare professionals

Externally initiated audits can be effective, but only, as reported above, if the audit is led by a champion, or if the healthcare professionals see the audit as a learning opportunity rather than as an external evaluation. This element was also recognised in a previous study into the barriers and facilitators of internal audits in preparation for an external audit: when healthcare professionals perceive internal audits as an examination tool that is only implemented because of external obligations they are less motivated to use the internal audit to drive improvements and their motivation will be lower during a follow-up audit [134]. Moreover, previous research has shown that involving healthcare professionals can be a critical factor for implementing and sustaining change. For example, a study by Hovlid et al. (2020) have showed that if organisations involve healthcare professionals in assessing care delivery for external audits, they are able to gain a better understanding of their current practice and consequently initiate action to improve quality of care. In organisations were involvement of healthcare professionals was low, no actual improvements were initiated or realized apart from updating written guidelines describing how care should be delivered [135].

### The audit as a platform for collaborative practice

Audits can also be initiated in a bottom-up way by intrinsically motivated healthcare professionals who collaborate with one another to improve their own local care practices. Previous research has highlighted the importance of a local, bottom-up, practice-based improvement approach [136, 137]. We add depth to these findings by showing that the strength of bottom-up QI comes from the fact that healthcare professionals are in the lead, share the same language and feel a shared ownership of the incremental changes, aspects that are lacking when having an audit imposed upon them [136].

### Different approaches to QI

In our review, we positioned audits as a QI approach to improve patient care and outcomes. Previous studies have often recommended a balanced or hybrid approach to audits in which organisations balance top-down control with empowering healthcare professionals to enable bottom-up improvements [138–140]. Surprisingly, we did not find any evidence to either support or reject a balanced or hybrid approach to audits. Despite this lack of evidence, we did observe that audits themselves are becoming increasingly hybrid and balanced, with different techniques and approaches being passed back and forth between different types of audits [38, 134]. This strengthens the claim that the CMOcs we identified are valid across the spectrum of audits.

### Is there such a thing as audit tiredness?

This review shows that with each year that an organisation participates in externally initiated audits, further improvements appear to tail off. While organisations initially invest heavily in order to satisfy the first accreditation visit, and maximise the benefits from the ensuing changes, after 3–10 years the learning curve levels off [83]. It seems that healthcare professionals no longer consider the audit to be a driver of change, and organisations then need to find

other initiatives to revive the process [3, 4, 6]. Whether a similar levelling off occurs with bottom-up initiatives led by the healthcare professionals themselves is uncertain, but it is certainly plausible that here also an audit 'tiredness' may ensue after some time.

## Practical implications

This study is of use to other researchers in this field, since it provides a framework for conceptualizing audits in their specific organisational context. The CMOcs that were identified offer policymakers and practice leaders an understanding of the mechanisms that promote the successful implementation of audit activities and of contextual factors that can either impede or support these mechanisms. Several ways to overcome the challenges facing audits are suggested, focusing on areas to consider when designing and optimising audit activities (Box 2). These recommendations are sufficiently generic to allow local tailoring to specific types of audits in varying contexts.

Our study suggests that healthcare needs to find additional ways to sustain the motivation and engagement of healthcare professionals in QI, as is also suggested by the fact that in 39% of papers sustainability as item in the QI-MQCS was lacking (S1 Table). Other, largely untested, approaches could include feedback and public reporting of patient-reported outcomes and experience measures (PROMs and PREMs) and the use of patient narratives and patient journeys to inform change and the redesign of care pathways [4, 141, 142], without any increase of the current administrative burden on health care professionals [143, 144].

---

### Box 2. How to make audits work effectively

1. Build the audit on teamwork and engage all healthcare professionals involved (identification of stakeholders). Utilise the shared commitment of the entire team working together to identify and spread effective practice. This can be achieved more easily within the local bottom-up initiated audits as professionals already recognise issues of interested from their own practice, share the same language and frame of reference and feel ownership of the incremental changes, rather than having an audit topic imposed on them.

2. Focus on knowledge-sharing within externally motivated audits. Healthcare professionals need to perceive the audit as a learning opportunity. This will only occur if attention and dedicated time is given to share knowledge with colleagues about the quality of care as it relates to the design of the care pathway. Preferably, this is blended with the approach described under 1.

3. Ensure there are local champions. If the driving force behind the audit is a local champion and peer, healthcare professionals are more likely to display ownership and see the audit process as worth the effort. Also, champions will challenge their colleagues to implement changes in practice.

4. Encourage during the audit the provision of feedback between healthcare professionals from all levels of the organisation. The audit needs to be positioned in such a way that all professionals feel it is safe to give feedback, also to someone who is perceived to by higher up the hierarchy.

---

This study is of use to other researchers, since it provides a framework for conceptualizing audits in their specific organisational context. We strongly urge authors of primary studies in this field to include detailed descriptions of the audits, the context in which they take place (including institutional, system and process aspects that could influence the adoption of audits), the audit process itself and how to sustain relevant outcomes in practice. This will permit a more fine-grained analysis of the mechanisms and contexts that impede or support the intended outcome.

## Limitations and strengths

This review has two fundamental limitations. First, most of the included articles report a positive response, which suggests a publication bias in that failed audits are likely not reported. Despite this possible bias, we were able to refine and advance our initial programme theories such that they can help to explain how audits might, or might not, lead to improved quality of hospital care.

Second, while this review has not examined interactions between the various CMOcs, they do seem to be interrelated. For example, those in leadership positions play an important role in ensuring a safe and trusting environment for the execution of audits, which in turn is a precondition for several of the mechanisms to come into play (i.e. for champions to play an active role in the audit, for giving and receiving feedback, for healthcare professionals to take ownership of the delivery of QI activities). Consequently, collecting primary data to explore these contextual factors and CMOcs would be an important step in further advancing our understanding of how and why audits might work. Also, we saw within our first CMOc (that externally initiated audits create QI awareness but their impact on improvement diminished over time) that successful audits can change the conditions that make them work in the first place. As such, CMOcs can be linked with the outcome of one phase of an audit becoming a contextual aspect for the next phase [145]. In this study, this ripple effect is premised on the idea that audit activities are "a series of events in the life course of a system, leading to the evolution of new structures of interaction and new shared meanings" [146] (p. 267).

In terms of strengths, this study contributes to the growing use of realist approaches in evidence synthesis. The realist approach is still developing, and key concepts are not always explained or applied in the same manner. For example, challenges have been identified in defining and operationalising mechanisms [24, 26, 27, 147, 148]. In this regard, Pawson (2012) observed that mechanisms "capture the many different ways in which the resources on offer may affect the stakeholders' reasoning" [149] (p.187). Given that resources and reasoning are both constituent parts of a mechanism, explicitly disaggregating them has helped to understand the ways in which mechanisms affect outcomes. By systematically applying methodological guidelines and describing our understanding of the key concepts, we have stuck closely to the realist synthesis approach [19]. We have provided a detailed account of our search methodologies and the process of theory elicitation and selection (S1 Protocol) [9]. We believe that these strategies have enhanced the transparency, and thus improved the validity, of our review.

## Conclusions

This realist review has indicated that champions are vital for an audit to be perceived by healthcare professionals as worth the effort. In addition, an audit can be an instrument that encourages healthcare professionals to provide each other with feedback, as well as to raise issues with leaders and become engaged in the governance of the organisation. As the broader learning- and QI infrastructure continues to mature, it will become increasingly important to think of audits in their wider context, especially one as complex and dynamic as a hospital care

system, when designing and optimising audit processes. We believe that this work on theorizing audits using a realist review methodology will provide policy makers and practice leaders with sufficient conceptual grounding to design contextually sensitive audits in a wide variety of settings. Future research could test the validity of our configurations through empirical studies that include detailed process evaluations of audits in order to provide further insights into the mechanisms and contextual factors through which audits produce their results.

## Supporting information

**S1 Protocol.**
(PDF)

**S1 Box. Search strategy.**
(DOCX)

**S1 Table. Characteristics of the studies included in the review.**
(DOCX)

**S2 Table. List of codes.**
(DOCX)

**S3 Table. PRISMA checklist.**
(DOCX)

## Acknowledgments

We wish to thank the participants of the focus group for their participation and insight. We would like to thank Johanna Schönrock-Adema for her constructive comments.

## Author Contributions

**Conceptualization:** Lisanne Hut-Mossel.

**Data curation:** Lisanne Hut-Mossel.

**Formal analysis:** Lisanne Hut-Mossel.

**Investigation:** Lisanne Hut-Mossel, Kees Ahaus, Gera Welker, Rijk Gans.

**Methodology:** Lisanne Hut-Mossel, Kees Ahaus, Gera Welker, Rijk Gans.

**Supervision:** Kees Ahaus, Gera Welker, Rijk Gans.

**Writing – original draft:** Lisanne Hut-Mossel, Kees Ahaus, Gera Welker, Rijk Gans.

**Writing – review & editing:** Lisanne Hut-Mossel, Kees Ahaus, Gera Welker, Rijk Gans.

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
