## [Decision Letter · Decision Letter 0]

11 Jan 2021

PONE-D-20-32155

Understanding how and why audits work in improving the quality of hospital care: a realist review.

PLOS ONE

Dear Dr. Hut-Mossel,

Thank you for submitting your manuscript to PLOS ONE. After careful consideration, we feel that it has merit but does not fully meet PLOS ONE’s publication criteria as it currently stands. Therefore, we invite you to submit a revised version of the manuscript that addresses the points raised during the review process.

I have reviewed the paper in addition to the reviewer. I am happy with its quality. However, please respond to the reviewers comments. 

We look forward to receiving your revised manuscript.

Kind regards,

Andrew Soundy

Academic Editor

PLOS ONE

Journal Requirements:

2. Please ensure that you have addressed all items recommended in the PRISMA checklist including identifying the study as a systematic review in the title. Please amend both the title on the online submission form (via Edit Submission) and the title in the manuscript so that they are identical.

3. In the methods, please clearly state your exclusion criteria.

4. Please include results and discussion of the quality of studies included in this review.

Reviewers' comments:

Reviewer's Responses to Questions

**Comments to the Author**

1. Is the manuscript technically sound, and do the data support the conclusions?

Reviewer #1: Yes

2. Has the statistical analysis been performed appropriately and rigorously? 

Reviewer #1: I Don't Know

3. Have the authors made all data underlying the findings in their manuscript fully available?

Reviewer #1: Yes

4. Is the manuscript presented in an intelligible fashion and written in standard English?

Reviewer #1: Yes

5. Review Comments to the Author

Reviewer #1: In this manuscript the authors report a systematic review of mechanisms and contextual factors underlying cinical audit, and their meta-narrative analysis of identified themes

I have some general comments

- On several occasions the authors discuss causality e.g. CMOcs leading to improved care quality and if-then programme theories. This is misleading. Given that the review is of observational studies, language should instead be observed associations and the authors' inferences

- PRISMA is referenced, but it's not clear that the manuscript follows guidelines. For example, title must specify that this is a systematic review

- I suspect that 'realist' methods may be unfamiliar to many. Suggest replace it in title (e.g. qualitative) and provide a brief basic explanatory paragraph (e.g. in methods) for the uninitiated

- You discuss audit, QI and quality assurance. It would be informative to include some examples and briefly highlight interactions/ differences/ overlap between the terms

- I'm unable to comment on the detail of realist synthesis methods

6. PLOS authors have the option to publish the peer review history of their article (what does this mean?). If published, this will include your full peer review and any attached files.

Reviewer #1: **Yes: **CM Oliver

---

## [Author Response · Author response to Decision Letter 0]

5 Feb 2021

Response to Reviewers PLOS ONE “Understanding how and why audits work in improving the quality of hospital care: a systematic realist review”

Thank you for your interest in our paper and for taking the time to express your feedback and suggestions. We have reviewed our paper according to the style requirements and have ensured that all submitted documents conform accordingly.

2. Please ensure that you have addressed all items recommended in the PRISMA checklist including identifying the study as a systematic review in the title. Please amend both the title on the online submission form (via Edit Submission) and the title in the manuscript so that they are identical.

Thank you for raising this point. We have chosen to use a realist review approach in addition to the approach of a systematic review. We’ve followed the RAMESES publication standards for realist reviews, which are based on the PRISMA statement, to ensure quality of reporting (see Wong et al. BMJ Open 2013). We agree with the editor and the reviewer that our approach meets the requirements of a systematic review as well. Therefore, we incorporated the term systematic realist review in the title of our manuscript, which now reads: “Understanding how and why audits work in improving the quality of hospital care: a systematic realist review”. 

3. In the methods, please clearly state your exclusion criteria.

During the development of our review protocol (see Hut-Mossel et al. BMJ Open 2017), we’ve formulated the inclusion criteria for this review (see Box 1 of this paper and the review protocol). Articles were excluded if they did not meet the inclusion criteria. We have now added extra information about the criteria for this review on page 7 and 8: 

“No changes were made to the inclusion criteria as formulated in the review protocol (S1 Protocol) [9]. Eligible studies were empirical studies that evaluated the effects of audits in hospital settings within high-income countries, and no restrictions were placed on the type of study design (Box 1). Articles were excluded if they did not meet the inclusion criteria [9]. Furthermore, articles reported in languages other than English were excluded to avoid misinterpretation of the content of an article due to language barriers.”

4. Please include results and discussion of the quality of studies included in this review.

Thank you for pointing this out. During our review, we’ve assessed the quality and the completeness of the included studies using two methods, namely the criteria established by the Cochrane ‘Effective Practice and Organisation of Care’ (EPOC) review group and by using the Quality Improvement Minimum Quality Criteria Set’ (QI-MQCS). More details about the quality of specific studies can be found in Supporting information 3. Characteristics of the studies included in the review. 

To further clarify how the quality assessment according to EPOC and QI-MQCS was interpreted in our study we have added more details in the results and discussion about QI-MQCS and its interpretation on page 8: 

“In order to interpret the QI-MQCS score, we used the criteria previously used by Kampstra et al. [32]: a study was weighed to be of perfect quality if > 15 items ranked yes, good quality if > 12 items ranked yes, moderate quality if > 9 items ranked yes) and insufficient quality if ≤9 items ranked yes.”

Page 9 and 10:

“Using the QI-MQCS, the completeness of reporting scores ranged from 4 to 16 (16 being the highest possible score) (S3 Table). Of the 85 included papers, 41 scored more than 12 items yes and where thus considered to be of good quality. 30 articles scored poorly on the QI-MQCS with a score ≤ 9, which is ranked as low quality. Only 61 of the 85 papers met the criteria for domain 12 (organisational readiness), that describes the QI culture and resources present in the organisation, which helps to assess the transferability of results. In addition, 52 of the 85 papers met the sustainability domain by including reference to organisational resources and policy changed needed to sustain the audit results after withdrawal of study personnel and resources. The synthesis began with evidence from the most rigorous studies (> 12 items ranked yes according to the QI-MQCS), as these studies provided the most relevant and detailed evidence about C, Mresource, Mreasoning or O. Less rigorous studies were used to test the emerging CMOcs, but also to test alternative explanations.”

Page 25

“Our study suggests that healthcare needs to find additional ways to sustain the motivation and engagement of healthcare professionals in QI, as is also suggested by the fact that in 39% of papers sustainability as item in the QI-MQCS was lacking (S3 Table).”

Reviewer 1:

General comments:

1. On several occasions the authors discuss causality e.g. CMOcs leading to improved care quality and if-then programme theories. This is misleading. Given that the review is of observational studies, language should instead be observed associations and the authors' inferences.

Thank you for your valuable feedback; it helps us to improve our explanation of and the reader’s understanding of CMOcs. Indeed, the initial programme theories (in the form of ‘If-then statements’) are quite conclusive and suggest causality, however, these theories were preliminary theory informed associations, which we have refined during the review. The development, testing and refinement of CMOcs in this systematic realist review provides an explanation of how and why audits might work and can, as such, be seen as observed associations. We have explained this more specifically in the definition of CMOcs. 

A strength of realist reviews is that the unit of analysis is not the intervention, but the mechanisms that underpin audits. For many of our CMOcs, we were able to draw on studies describing different types of audits that shared the same programme theory. Each CMOc was grounded by several papers about different types of audits which pointed in the same direction. We’ve added further clarification in Table 2. Glossary of terms on page 6 & 7:

Realist review – Is a theory-driven approach to synthesising quantitative, qualitative or mixed methods research, from a perspective based in Realism. One of the key tenets of realist methodology is that programmes work differently in different contexts – hence an audit that achieves ‘success’ in one setting may ‘fail’ (or only partially succeed) in another setting, because the mechanisms needed for success are triggered to different degrees in different contexts [25,28]. A realist review does not provide a definite answer as to whether ‘something works or not’; rather it answers questions of the general format ‘what works for whom under what circumstances, how and why?’ to enable informed choices about further use and/or research. The results of a realist explanation are context-mechanism-outcome configurations (CMOcs) constituting a refined programme theory. The refined theory can then be tested in a subsequent realist evaluation covering the same kind of project or programme.

Context – Context refers to the ‘setting’ in which interventions take place. As conditions change over time while the audit is executed, the context may change as well and reflects aspects of that change in conditions. As such, contextual elements influence the relationship between audits and their outcomes and, vice versa, the process of the audit and its outcome will influence the context (for example, the outcomes of an audits may generate a culture change) [24,25]. 

Mechanism – Mechanisms are a combination of resources offered by the intervention (Mresource e.g. information, skills or support) and the participants’ response to these resources (Mreasoning). Intervention resources are introduced in a context that might affect participants behaviour based upon their beliefs, attitudes or logic (reasoning), which will, in turn, affect the outcome [24,25,27]. 

Outcome – Outcomes can be either intended (did the project succeed against the criteria it set itself at the outset), or unintended, and can be proximal or final. Proximal outcomes can be changes in skills, commitment or intentions of healthcare professionals, whereas final outcomes relate to improvements in quality of care.

Context-mechanism-outcome configuration (CMOc) – In realist reviews, observed associations are expressed as context-mechanisms-outcome (CMO) configurations, to explain how particular contexts trigger mechanisms to generate certain outcomes [24]. CMO configurations can be applied to design future audits across different contexts.

Programme theory – Programme theories are theory informed associations organised as ‘abstracted descriptions’ about the ideas and assumptions underlying how, why and in what circumstances complex social interventions work [19]. Typically, realist reviews start with initial programme theories (in our review formulated as ‘If-then statements’) and end with a more refined programme theory, expressed as CMOcs. 

In addition, we’ve nuanced the paragraph about limitations and strengths on page 25:

“This study is of use to other researchers, since it provides a framework for conceptualizing audits in their specific organisational context. We strongly urge authors of primary studies in this field to include detailed descriptions of the audits, the context in which they take place (including institutional, system and process aspects that could influence the adoption of audits), the audit process itself and how to sustain relevant outcomes in practice. This will permit a more fine-grained analysis of the mechanisms and contexts that impede or support the intended outcome.”

2. PRISMA is referenced, but it's not clear that the manuscript follows guidelines. For example, title must specify that this is a systematic review.

Thank you for addressing this point. We have added the PRISMA checklist as Supporting information 5 and described the PRISMA statement now more explicitly in the methods on page 5:

“We adopted a realist review approach to address our research questions. We were guided by the RAMESES publication standards for realist reviews and we followed the PRISMA guidelines for systematic searching of the literature (S5 Table) [19,20].”

Furthermore, your suggestion to typify our realist review as a systematic review was also expressed by the editor. We agree with both of you that our approach meets the requirements of a systematic review as well. Therefore, we have incorporated the term systematic realist review in the title of our manuscript, which now reads: “Understanding how and why audits work in improving the quality of hospital care: a systematic realist review”.

3. I suspect that 'realist' methods may be unfamiliar to many. Suggest replace it in title (e.g. qualitative) and provide a brief basic explanatory paragraph (e.g. in methods) for the uninitiated.

You have raised an important point. Realist review is a new approach, however it is growing in popularity and becoming much more established as an approach in health research. We have noted that, also in PLOS ONE, more realist reviews are published recently, for example the systematic realist review by Sud et al. (PLOS ONE, 2020). One of the key tenets of realist methodology is that programmes work differently in different contexts – hence an audit that achieves ‘success’ in one setting may ‘fail’ (or only partially succeed) in another setting, because the mechanisms needed for success are triggered to different degrees in different contexts. We hope that PLOS ONE readers are growing in familiarity with the realist approach and that our systematic realist review will help to add to this understanding. We have added referencing to table 2. Glossary of terms on page 5 and provided a definition of realist review into table 2. Glossary of terms on page 6 & 7:

 “Table 2 provides definitions of realist concepts.”

“Realist review – Is a theory-driven approach to synthesising quantitative, qualitative or mixed methods research, from a perspective based in Realism. One of the key tenets of realist methodology is that programmes work differently in different contexts – hence an audit that achieves ‘success’ in one setting may ‘fail’ (or only partially succeed) in another setting, because the mechanisms needed for success are triggered to different degrees in different contexts [26,29]. A realist review does not provide a definite answer as to whether ‘something works or not’; rather it answers questions of the general format ‘what works for whom under what circumstances, how and why?’ to enable informed choices about further use and/or research. The results of a realist explanation are context-mechanism-outcome configurations (CMOcs) constituting a refined programme theory. The refined theory can then be tested in a subsequent realist evaluation covering the same kind of project or programme.” 

And added extra information about why we have chosen to use a realist review approach on page 5:

“In addition to the approach of a systematic review, we have chosen to use a realist review approach because this permits us to understand in what circumstances and through what processes audits might, or might not, lead to improved quality of hospital care and why. This approach recognises that the success of audits is shaped by the way in which they are implemented and the contexts in which it is implemented.”

4. You discuss audit, QI and quality assurance. It would be informative to include some examples and briefly highlight interactions/ differences/ overlap between the terms

Thank you for pointing this out. It is indeed relevant to elaborate on audit, QI and QA. We have added Table 1. Types of audits on page 3 & 4 to clarify the distinction between these terms:

Externally driven audits – For example, accreditation, certification and external peer reviews are strongly anchored in quality assurance (QA), referring to initiatives designed to assure compliance with minimum quality standards [10,11]. Quality assurance is defined as: “The part of quality management focused on providing confidence that quality requirements will be fulfilled” [11]. External audits are used to assess the quality system of a healthcare organization based on specified standards and are conducted by external auditors [12].

Internal audits – This type of audit is conducted by internal auditors of the hospital’s own organisation, such as quality officers or healthcare professionals from another department than the one being audited to guarantee independent judgement. Internal audits are used to evaluate the quality system based on standards as well. They are conducted to prepare for external audits. Healthcare organisations also use internal audits to continuously improve the quality of healthcare. Internal audits are designed to evaluate and improve the effectiveness of the organisation’s quality management system and focus more on organisational conditions than on performance of healthcare professionals and patient outcomes [3].

Clinical audits – Clinical audits differ from other types of audits in that they are mostly initiated and undertaken by healthcare professionals. Clinical audits represent a shift from QA to QI, with a focus on increasing the ability to fulfil quality requirements, seeking to improve care, enhance performance and prevent poor care [11,13]. This process takes place continuously as part of everyday routines [1,12,13]: healthcare professionals work together to collect data and evaluate their own practices. Following this, they intend to develop and apply improvements in daily practice, and then the audit cycle is repeated to demonstrate improved and sustained improvements [14]. Hence, clinical audits do not necessarily use external criteria and are not carried out in response to external demands as the initiative comes from the healthcare professionals themselves [15].

5. I'm unable to comment on the detail of realist synthesis methods

We hope that with the additional, properly referenced information sufficient detail of realist synthesis methods have been provided.

---

## [Editor Report · Decision Letter 1]

4 Mar 2021

Understanding how and why audits work in improving the quality of hospital care: a systematic realist review.

PONE-D-20-32155R1

Dear Dr. Hut-Mossel,

We’re pleased to inform you that your manuscript has been judged scientifically suitable for publication and will be formally accepted for publication once it meets all outstanding technical requirements.

Kind regards,

Andrew Soundy

Academic Editor

PLOS ONE
---

## [Editor Report · Acceptance letter]

15 Mar 2021

PONE-D-20-32155R1 

Understanding how and why audits work in improving the quality of hospital care: a systematic realist review  

Dear Dr. Hut-Mossel:

I'm pleased to inform you that your manuscript has been deemed suitable for publication in PLOS ONE. Congratulations! Your manuscript is now with our production department. 

Kind regards, 

on behalf of

Dr. Andrew Soundy 

Academic Editor

PLOS ONE